# Fault Detection and Diagnosis in Industry 4.0: A Review on Challenges and Opportunities

**DOI:** 10.3390/s25010060

**Published:** 2024-12-25

**Authors:** Denis Leite, Emmanuel Andrade, Diego Rativa, Alexandre M. A. Maciel

**Affiliations:** 1Mekatronik I.C. Automacao Ltda, Rua Sargento Silvino Macedo, 130—Imbiribeira, Recife 51160-060, PE, Brazil; 2Instituto de Inovação Tecnológica—IIT, Universidade de Pernambuco—UPE R. Min. Mario Andreaza, s/n—Várzea, Recife 50950-050, PE, Brazil; emmanuel.andrade@poli.br (E.A.); diego.rativa@poli.br (D.R.); alexandre.maciel@upe.br (A.M.A.M.)

**Keywords:** fault detection, fault diagnosis, intelligent manufacturing systems, machine learning, smart manufacturing

## Abstract

Integrating Machine Learning (ML) in industrial settings has become a cornerstone of Industry 4.0, aiming to enhance production system reliability and efficiency through Real-Time Fault Detection and Diagnosis (RT-FDD). This paper conducts a comprehensive literature review of ML-based RT-FDD. Out of 805 documents, 29 studies were identified as noteworthy for presenting innovative methods that address the complexities and challenges associated with fault detection. While ML-based RT-FDD offers different benefits, including fault prediction accuracy, it faces challenges in data quality, model interpretability, and integration complexities. This review identifies a gap in industrial implementation outcomes that opens new research opportunities. Future Fault Detection and Diagnosis (FDD) research may prioritize standardized datasets to ensure reproducibility and facilitate comparative evaluations. Furthermore, there is a pressing need to refine techniques for handling unbalanced datasets and improving feature extraction for temporal series data. Implementing Explainable Artificial Intelligence (AI) (XAI) tailored to industrial fault detection is imperative for enhancing interpretability and trustworthiness. Subsequent studies must emphasize comprehensive comparative evaluations, reducing reliance on specialized expertise, documenting real-world outcomes, addressing data challenges, and bolstering real-time capabilities and integration. By addressing these avenues, the field can propel the advancement of ML-based RT-FDD methodologies, ensuring their effectiveness and relevance in industrial contexts.

## 1. Introduction

The rise of Industry 4.0 has brought about significant changes in manufacturing, with Artificial Intelligence (AI) playing a crucial role in enhancing operational performance [1]. AI goes beyond simple automation, providing valuable insights and analytics that help industries optimize their processes, minimize downtime, and increase efficiency. According to a report by the World Economic Forum on AI in Manufacturing, AI applications have significantly improved productivity, quality, and flexibility in industrial settings. This indicates a transformation in how manufacturing processes are managed and executed [1].

In this context, Fault Detection and Diagnosis (FDD) is a key industrial automation and control engineering subdomain. Fault detection involves identifying anomalies or deviations from normal system behavior. In contrast, fault diagnosis focuses on isolating and determining the root cause of these anomalies [2]. These processes enable timely interventions, which help to reduce unplanned downtime and maintenance costs. The capability to execute FDD in real time (RT-FDD) is particularly vital in Industry 4.0 contexts, where real-time insights are essential for maintaining optimal production flow and preventing cascading failures [3,4].

In the industrial landscape, processes can be categorized into two main classes: continuous and discrete. Continuous processes, such as oil refineries, operate without interruptions and maintain a consistent level of product output. This steady-state operation leads to relatively stable and predictable process conditions, where deviations from the norm are gradual and often associated with predictable wear or inefficiencies. Therefore, RT-FDD approaches for continuous processes focus on detecting subtle, slow-developing anomalies in these stable conditions [5].

On the other hand, Discrete Manufacturing Machines (DMMs) represent examples of discrete processes that operate in distinct sequences to produce individual items. DMMs are characterized by intermittent start–stop operations with varying conditions, and each cycle can introduce unique, transient states, making fault detection more challenging due to the dynamic and varied nature of the operation. Consequently, RT-FDD in DMMs must be adaptable to rapidly changing conditions and capable of interpreting complex and transient data patterns. This starkly contrasts the more stable and predictable environment of continuous processes, where gradual trend analysis and long-term data monitoring are practical [4].

DMMs often include knowledge-based (KB) fault diagnostic systems that rely on sensors and parameters for alarm initiation [5]. However, KB diagnostics have inherent limitations, especially when dealing with the diverse and complex data types in DMMs. These methods require extensive human input to manually describe potential faults or anomalous situations, limiting detection to faults known and described by experts, and imposing constraints based on project scope, time, and cost. As a result, KB systems often need to recognize unforeseen or novel fault conditions, limiting their effectiveness in dynamic industrial environments [6,7]. These challenges require alternative methods to be explored, such as Physical Models (PM) and data-driven techniques like Machine Learning (ML), which offer more robust and comprehensive fault detection capabilities. Data-driven approaches, in particular, can learn from operational data to identify both known and previously unseen fault patterns, enhancing the system’s adaptability to a broader range of conditions and reducing dependency on expert-defined rules [8,9,10].

Further research on ML for RT-FDD in discrete manufacturing systems, such as DMMs, is essential for advancing the goals of Industry 4.0 and the digital transformation journey. Most existing studies have focused on continuous processes, and the industry’s limited expertise in ML presents a significant challenge [11,12,13,14,15,16]. One of the major barriers to widespread ML adoption in the industry is the need for more data scientists, compounded by issues of interpretability and the absence of benchmark datasets [17,18,19,20,21,22,23,24]. While researchers have made commendable contributions, the field lacks a unified trajectory due to fragmented research paths and a scarcity of sustained, directional efforts. However, it is worth noting that the number of studies in this area has increased significantly, with fourteen new studies emerging in 2022 and 2023, indicating that this area is becoming a hot research topic due to its big impact on the manufacturing industry.

In this manuscript, we have conducted a Systematic Literature Review (SLR) to synthesize insights from prior studies and highlight implementation challenges and research opportunities. The focus of this review is on bridging the gap between theoretical advancements and their practical implementation, which is crucial for realizing the full potential of Real-Time Fault Detection and Diagnosis (RT-FDD) in industrial applications. We have proposed seven Literature Review Research Questions (LRRQ) to help guide readers through the main challenges and benefits of implementing ML-based RT-FDD in the industry and future research opportunities.

The manuscript is structured as follows: Section 2 explores the FDD fundamentals, outlining the characteristics and challenges of continuous processes and discrete manufacturing systems while contrasting classical and data-driven approaches. Section 3 presents a comprehensive Systematic Literature Review (SLR), categorizing existing studies into six thematic groups and highlighting the problems, methodologies, and challenges addressed. Section 4 discusses the findings, identifies key challenges, and outlines future research opportunities, such as standardized datasets, Explainable AI, and improved feature extraction techniques, which have the potential to inspire and motivate further research. Section 5 integrates the insights from the review, synthesizing contributions and actionable recommendations to bridge gaps in research and practice. Finally, Section 6 concludes the paper by summarizing key findings, emphasizing the critical importance of collaboration between technological advancements and industrial application, and offering a roadmap for advancing RT-FDD in the context of Industry 4.0.

## 2. Fault Detection and Diagnosis in the Industrial Domain

An industrial process is a series of interconnected tasks or operations to produce a specific product or service. The type of industrial process used is characterized by the product’s nature, the production volume, and the level of automation involved. These processes are integral to optimizing resource utilization, reducing waste, and enhancing overall manufacturing efficiency [25].

Continuous processes are a type of industrial process where raw materials or inputs are continuously fed into the system, resulting in a constant stream of finished products. These processes are typically automated, operate 24/7, and are suitable for large-scale production of chemicals, petroleum, and other commodities. The consistent nature of continuous processes enables better control over product quality and production rates [26]. In continuous processes, rotary machines are preferred for their ability to handle a continuous flow of materials efficiently.

Batch processes produce goods in finite quantities, where production occurs in separate batches or runs. This form of processing is common in the pharmaceutical, food, and specialty chemical industries, where the product variety is high, and customization is essential. Batch processes allow for flexibility in production, but may require more significant human intervention and longer downtime between batches, according to the same source [26]. Discrete sequential cyclic machines are used to process materials or products in distinct batches, allowing for flexibility in product variation and production scheduling.

Cyclic processes are characterized by repeating a set of operations or tasks in a specific order, with the cycle repeating after the final operation. This type of process is common in assembly lines and machining, where products undergo a series of operations in a predetermined sequence. Cyclic processes balance flexibility and efficiency, making them suitable for various manufacturing scenarios [26]. Depending on the specific tasks involved, cyclic processes may combine rotatory and discrete sequential cyclic machines in the production sequence.

Industrial rotary machines usually include rotating parts like shafts and rotors powered by an electric motor [27]. In terms of static behavior, these machines operate at a steady state when they attain a stable rotational speed and produce a constant output [28]. The static behavior of these machines can be characterized by parameters such as torque, power, and efficiency [27]. Regarding dynamic behavior, industrial rotary machines exhibit rotational motion and can experience vibrations and oscillations during operation [29]. The dynamic behavior of these machines can be characterized by parameters such as natural frequencies, damping ratios, and vibration amplitudes [29].

On the other hand, discrete sequential cyclic machines have more intermittent operations with start–stop cycles for each operation. These machines involve a series of discrete movements and transitions between operations, which can also result in dynamic behavior such as acceleration and deceleration [30]. The static behavior of these machines can be measured by parameters such as cycle time, throughput, and utilization [31]. In this case, the dynamic behavior can be measured by parameters such as acceleration/deceleration rates, jerk, and positional accuracy [32].

Consequently, anomalous conditions or faults are expected to be directly related to the control variable values. For instance, in electric motors, current, voltage, and temperature measures can be monitored to detect abnormalities that may indicate faults or malfunctions [27]. Similarly, in stamping presses, force, displacement, and vibration measures can be monitored to detect issues such as misalignment, wear, or excessive load [31]. In discrete sequential cyclic machines, measuring position, velocity, and acceleration can help detect deviations from the expected behavior and identify potential faults or anomalies [30].

### 2.1. Industrial Automation Systems

Industrial automation systems play a vital role in improving the efficiency of manufacturing and production facilities. Automation involves the use of control systems to operate machinery and equipment without the need for human intervention. Control refers to the ability to manage and regulate the functions of a system through direct commands or programmed settings, ensuring that it performs as intended, typically with feedback systems in place to adjust actions based on monitored variables [33].

Industrial automation systems consist of several key components, including Programmable Logic Controllers (PLCs), Input/Output devices (IOs), and Supervisory Control and Data Acquisition (SCADA) systems. PLCs are digital computers that control manufacturing processes, such as assembly lines or robotic devices [34]. They are versatile, and are essential in real-time decision-making and communication within the system. IOs serve as the interfaces through which PLCs receive sensor data and transmit control signals to actuators. Meanwhile, SCADA systems are highly configured systems used to control and monitor facilities and infrastructure, which provide a graphical user interface to observe and interact with the entire system [35].

Many industrial automation systems use alarm systems incorporated within PLCs, which are based on predefined rules and help operators quickly identify and diagnose faults. However, these alarm systems have a significant drawback: they are limited by the human ability to create these rules. As a result, they can only identify and address a limited number of known situations, potentially missing rare or unexpected anomalies [36]. Therefore, the need for more advanced and adaptive Fault Detection and Diagnosis mechanisms is highlighted by the dependence on rule-based alarm systems, which are limited by human foresight.

### 2.2. Foundational Fault Detection and Diagnosis Approaches

Faults can harm industrial processes and machinery, resulting in significant operational and economic repercussions [37,38]. Fault Detection and Diagnosis (FDD) is a key subdomain in control engineering, combining principles from various disciplines to identify and rectify abnormal system behaviors. Fault detection involves identifying anomalies in a system’s operation. In contrast, fault diagnosis involves monitoring the system to isolate the source of the anomaly and determine its nature [2].

As represented in Figure 1, there are three main types of fault detection methodologies: knowledge-based, model-based, and data-driven. Knowledge-based methods use expert knowledge and rule-based systems to identify and diagnose faults. These systems employ heuristic rules or algorithms derived from human expertise in the specific domain, allowing them to make decisions or trigger alarms based on specific conditions or symptoms [39]. Model-based methods rely on mathematical models that represent the system’s normal behavior. This approach analyzes discrepancies between the model predictions and actual system outputs to detect faults. These methods are effective in systems where accurate models can be developed to simulate normal operations [40]. Data-driven methods, on the other hand, are well-suited for situations where developing explicit knowledge-based or model-based descriptions is prohibitively complex or resource-intensive. These methods use Machine Learning and statistical techniques to analyze large volumes of operational data, extracting patterns that indicate faults without requiring prior physical or heuristic models [41,42].

Figure 1 illustrates the intersection between industrial and computer science domains within the FDD domain. Knowledge-based and model-based approaches, grounded in industrial knowledge, utilize structured rules and physical models, making them more intuitive for domain experts. Contrarily, data-driven methods align with computer science expertise, employing advanced computational tools to extract insights from extensive datasets. Moreover, the figure highlights the continuous and discrete process characteristics, emphasizing the need for tailored FDD approaches to address their distinct characteristics effectively.

Therefore, recognizing the unique requirements of various process types, alongside the strengths and limitations of knowledge-based, model-based, and data-driven methodologies, is essential for developing a practical FDD framework and exploring these complementary approaches.

### 2.3. Data-Driven Fault Detection and Diagnosis

Transitions from classical FDD methodologies to data-driven approaches present challenges due to the complexity of specific systems, where traditional methods may be impractical due to limitations in modeling or available expertise [41,42]. Data-driven techniques leverage AI and Machine Learning to extract insights directly from data, complementing traditional methods with adaptability and scalability. This evolution highlights the transformative role of AI and ML in Fault Detection and Diagnosis, as detailed in the following section.

AI involves creating systems that possess human-like cognitive functions such as learning, reasoning, and problem-solving [43]. Its application ranges widely from autonomous vehicles to healthcare diagnostics [43]. On the other hand, ML is a crucial subset of AI that enables machines to learn from data, thereby revolutionizing the way complex problems are tackled [44]. It includes techniques like logistic regression and Neural Networks, which are essential in various industries, including fault detection and process optimization [45].

Data-driven methods are becoming increasingly popular in fault detection. These methods use historical process measurement data and Machine Learning techniques to learn the characteristics of faults and detect anomalies. By analyzing patterns in the data, these methods can identify irregularities that indicate potential faults. This is especially useful in complex systems where modeling may need to be improved or expert knowledge made available [13]. Data-driven fault detection methods have experienced substantial development in recent years, using process measurement data to discern fault characteristics through techniques such as multivariate statistical analysis and Machine Learning [13,46]. The increasing computational capabilities and availability of large datasets have further propelled the growth and applicability of data-driven methods in various industrial settings [47].

Emerging technologies and techniques in Artificial Intelligence, Machine Learning, and data analytics continue to shape the landscape of FDD, offering new possibilities and challenges in tackling complex industrial faults [43]. Some essential ML learning methods include supervised learning, where algorithms are trained on labeled data, which is crucial for predictive modeling in industrial settings [43]. On the other hand, unsupervised learning focuses on identifying patterns in unlabeled data and is used for anomaly detection. Lastly, reinforcement learning entails training models through trial and error and is significant for autonomous systems in dynamic environments. These methods are outlined in an itemized list referencing relevant literature [43,45].

Fault detection using ML has great potential in industrial settings. However, the current workforce needs more ML professionals, such as data scientists [48,49]. Researchers are developing Automated Machine Learning (AutoML) to bridge this gap, which allows non-ML experts to use ML technologies. AutoML automates the model development process, including feature engineering, model selection, and hyperparameter tuning, making it easier to apply ML in various domains, especially in those where ML expertise is limited [44,50]. This democratizes access to ML technologies, enabling interdisciplinary solutions and facilitating collaboration between domain experts and ML practitioners [18].

### 2.4. Previous Reviews

We conducted a thorough analysis of previous Systematic Literature Reviews and surveys with a focus on their primary objectives, key findings, and areas where further research is needed. To select the relevant studies, we searched for papers containing fault AND (detection OR diagnosis) AND (“machine learning” OR “artificial intelligence”) AND (review OR survey) in the title, in Science Direct, SCOPUS, and IEEE. A total of 33 studies have been recovered, from which 26 are focused on FDD in rotary machines, and 2 are short papers. Table 1 summarizes the remaining five reviews organized by the historical progression. Although the reviews on rotary machines have extensively explored their specific dynamics and challenges, including those reviews in our current analysis does not significantly contribute to our unique objectives.

In the 1990s, Patton conducted a comprehensive review of the Fault Detection and Diagnosis (FDD) theme [51]. The review highlighted the focus of that period on creating residual signals that could capture discrepancies between faulty and fault-free system operations. One of the significant challenges in their review was establishing suitable thresholds for these residual signals, which were crucial for distinguishing between normal and abnormal system behaviors. Furthermore, they provided definitions for terms such as fault diagnosis, which refers to the ability to pinpoint the specific fault present in a system; fault detection and isolation (FDI), which signifies the process of monitoring and identifying when a faulty condition arises; and fault-tolerant control, which represents the capability to generate corrective control actions in response to an identified fault.

Fenton et al. [52] conducted a review of fault diagnosis techniques for electronic systems that utilize intelligent methods. The analysis included rule-based (RB), model-based (MB), and case-based approaches, examining their applications. The review found that RBS, which has been used since the 1970s, is appreciated for its intuitive simplicity. However, it heavily relies on expert knowledge, struggles with novel faults, and is sensitive to system changes. Over the past 15 years, MB approaches have gained research prominence, but RB remains dominant in industrial applications. Various MB approaches include fault, behavioral, and diagnostic inference models. Fault models, for instance, are hierarchical systems built by simulating faults for early detection and isolation. While adequate for combinational digital circuits, their efficacy diminishes with sequential circuits. The authors emphasize the need for automated diagnostics in electronic systems, given the fast time-to-market, reduced product lifecycle, and heightened complexity. Still, despite research progress, solutions must be improved to suit real industrial applications, especially in deploying valuable tools that promote clear savings; otherwise, acceptance will be difficult.

Lo et al. [53] summarized the research on fault diagnosis in industrial systems using Machine Learning tools published from 2002 to 2018. They present the primary ML approaches applied to fault diagnosis in the industry, and discuss issues such as “data quality” (missing or noisy data), the ability to consider several types of variables (discrete or continuous). Furthermore, they explore the dynamic system’s temporal regime and the capacity of the models’ generalization. The review exposes that although ML is promising for fault diagnosis, only one method addresses some challenges in developing suitable models. Finally, they conclude that hybrid methods and approaches could be considered.

Fernandes et al. [54] performed a Systematic Literature Review of the Machine Learning methods used to detect mechanical faults and the prognosis of faults in manufacturing equipment in real-world scenarios and presented empirical results from industrial case studies. To their knowledge, no prior systematic review on this specific topic focused on FDD of mechanical faults. Their investigation included the algorithms and methods currently deployed, their limitations and advantages, and which ones are used for data stream learning.

According to the authors, while each study has focused on the detection of mechanical faults or the prognosis of faults in real manufacturing scenarios, they differ significantly in three essential aspects: the manufacturing context in which the study is undertaken, the machinery for which faults were detected or predicted, and the characteristics of the available data. Although expected in industrial case studies, these differences have made it challenging to compare the different techniques. Moreover, they highlight a significant issue that requires consideration when conducting fault detection and prognosis in the manufacturing industry: the inherent complexity of manufacturing systems and the time-varying properties of production processes. The review recommends further research into developing Machine Learning algorithms and methods capable of handling noisy, non-stationary data, and identifying nonlinear interaction patterns between machinery components. In particular, the importance of pursuing research in online learning is emphasized due to its capacity to continuously learn from new data generated by machines.

Finally, and no less important, the authors confirmed that the industry’s shortage of labeled data confines the learning task to unsupervised and semi-supervised methods. They conclude that it is imperative to demonstrate the effectiveness of these methods in real-world applications. Concerning the application field of predictive maintenance, meeting technical and economic requirements, and justifying investments requires careful consideration of aspects such as the models’ effectiveness, the computational power required, and the interpretability of the models.

The review by Arpitha et al. [4] offers a comprehensive exploration of Machine Learning and Deep Learning methods used in Fault Detection and Diagnosis within the metal etching process in semiconductor manufacturing. The authors categorize various techniques, such as those based on PCA, GMM, kNN, SVDD, and Deep Learning. Each category is scrutinized for its efficacy in addressing challenges like data nonlinearity, multimodality, and high dimensionality inherent in metal etching. Importantly, the review points out the limitations of traditional univariate techniques in this multivariate process context. By evaluating the pros and cons of each method and discussing their industrial applications, the review not only serves as an informative resource but also paves the way for future research in process monitoring and fault diagnosis in the semiconductor sector.

This historical overview, as summarized in Table 1, not only traces the evolution of methodologies in RT-FDD, but also reflects on the challenges and opportunities at each stage, contributing to the ongoing advancement of the field. The progression highlighted in these studies indicates a continuous refinement of techniques and approaches in RT-FDD. However, Fernandes et al. [54] have pointed out that diverse industrial contexts and non-uniform datasets make it difficult to compare studies. This fragmentation in research focus and objectives is also evident in the predominance of research on rotary machines, where numerous benchmark datasets exist [55,56,57,58,59,60,61,62,63,64,65,66,67,68,69,70,71,72,73,74,75,76,77,78,79,80].

In light of these findings, the research landscape in RT-FDD appears segmented, and there is a significant need for cross-industry benchmark datasets to establish a unified state of the art. However, this gap presents an excellent opportunity for further research, particularly in developing and sharing diverse datasets and assessing the applicability of Machine Learning techniques in real-world industrial settings. Identifying the barriers to broader implementation of these technologies and exploring new research directions is crucial for advancing RT-FDD.

The following section conducts a Systematic Literature Review (SLR) to evaluate the current state-of-the-art Machine Learning for RT-FDD and its contributions to the industry. The focus is on bridging the divide between theoretical advancements and their practical implementation, a key to realizing RT-FDD’s full potential in industrial applications.

## 3. Systematic Literature Review

The search is systematically conducted across four digital libraries with extensive coverage of information technology and engineering literature, following the protocol presented by Kitchenham [81] for conducting a Systematic Literature Review:IEEE Xplore for its wide range of electrical and electronic engineering publications.ACM Digital Library for its comprehensive collection of computing and information technology research.SCOPUS for its broad interdisciplinary coverage and citation data.Science Direct for its extensive repository of scientific and technical research.

Based on the reviews in the previous section, seven Literature Review Research Questions (LRRQ) have been developed. These questions guide our Systematic Literature Review (SLR) and cover various aspects of Real-Time Fault Detection and Diagnosis (RT-FDD). Our research goals align with these questions, aiming to ensure a comprehensive review.

**LRRQ1:** *What is the state of the art in RT-FDD?* Establishing the current benchmark for RT-FDD is essential in tracking its progress and setting a baseline for future innovation.**LRRQ2:** *Which research topics have been addressed in RT-FDD primary studies?* This question aims to identify the research topics covered by primary studies in order to ensure that the review covers the entire field spectrum and identifies any potential research gaps.**LRRQ3:** *Which ML techniques have been employed for RT-FDD?* Exploring the applied ML techniques provides insights into the technological advancements of RT-FDD, evaluating both the diversity and progression of computational approaches.**LRRQ4:** *What are the main challenges to implementing ML-based RT-FDD in the industry?* Identifying these challenges is essential to develop solutions, directing research efforts towards overcoming practical barriers to Machine Learning adoption in industrial Fault Detection and Diagnosis.**LRRQ5:** *What are the potential benefits of implementing ML-based RT-FDD in the industry?* Elucidating the value proposition of ML in RT-FDD by outlining its potential benefits can motivate stakeholders to invest in this technology.**LRRQ6:** *How has RT-FDD been addressed in continuous process and discrete manufacturing?* By examining RT-FDD applications in varied manufacturing processes, this question seeks to differentiate how methodologies adapt to and are effective in distinct industrial settings.**LRRQ7:** *What are the research opportunities?* This question is intended to inspire new studies that expand the boundaries of RT-FDD research.

In order to include all relevant studies, a search query is created with specific parameters aligned with the research questions. These parameters are adjusted based on the search syntax of each library to overcome their limitations. The primary goal of answering these questions is to better understand the use of ML in the context of FDD. By addressing these questions, we aim to achieve a deeper and more integrated understanding of ML in the context of FDD, ultimately leading to the development of more intelligent and reliable industrial systems.

To guarantee that the literature review covers all relevant aspects of the subject matter, we have carefully chosen the search parameters A, B, C, and D, which directly correspond to the key aspects of RT-FDD being investigated, as summarized in Table 2. By strategically selecting and combining these parameters, we can guarantee that our search query is both comprehensive and intricately connected to the core research questions. This will facilitate a thorough literature review that is aligned with the study’s overarching goals.

Parameter A focuses on the “Application Field,” and ensures that the literature covers industrial contexts where Real-Time Fault Detection and Diagnosis (RT-FDD) is most relevant. It addresses LRRQ2 and LRRQ4, which inquire about the thematic scope and application in manufacturing processes. On the other hand, Parameter B, ‘Application Purpose’, covers the various activities within RT-FDD, echoing LRRQ3’s probe into ML techniques and LRRQ5’s inquiry into implementation challenges.

Parameter C, “Computational Technique,” captures the breadth of ML approaches and defines the state-of-the-art. It is aligned with LRRQ3 and LRRQ1. Finally, Parameter D, “Time Aspect,” highlights the importance of immediacy in fault detection, which is crucial for answering LRRQ5 and LRRQ6 regarding the challenges and benefits of implementing ML-based RT-FDD in real-time scenarios.

The main aspects of our study are represented by a Venn diagram with four overlapping circles shown in Figure 2. These aspects include Industrial Processes or Machinery, Machine Learning, Fault Detection and Diagnosis, and Real Time. The central region, marked as ‘R’, represents the specific focus of our review, which is the application of Real-Time Fault Detection and Diagnosis (RT-FDD) using Machine Learning techniques in industrial settings.

The ‘R’ region has multiple aspects relevant to the literature review research questions. For instance, it directly informs LRRQ2, which inquires about the Machine Learning techniques used for RT-FDD, by focusing on studies that employ these techniques in a real-time, industrial context. The ‘R’ region also helps determine the state of the art (LRRQ3) by concentrating on the latest and most relevant studies that combine these four dimensions. The challenges and potential benefits of implementing ML-based RT-FDD in industry, as outlined in LRRQ4 and LRRQ5, are inherently tied to the real-time operational requirements and the industrial application of these systems. Lastly, the diagram and the corresponding ‘R’ region provide a visual and conceptual tool for identifying gaps and research opportunities (LRRQ6) by highlighting the convergence of ML in real-time fault detection within industrial processes.

### 3.1. Inclusion and Exclusion Criteria

The primary focus is to include original research articles directly addressing FDD or anomaly detection within industrial machinery or processes, ensuring a concentration on studies that provide innovative insights or propose novel methods in the field of FDD, leveraging Machine Learning (ML) and related technologies. On the other hand, the exclusion criteria are carefully chosen to omit studies that do not align with the core focus of this review. For instance, studies that are narrowly focused on specific sensing technologies or specific types of machinery and those that do not directly deal with ML in the context of FDD are excluded.

Following the initial and secondary screening, the final selection of articles is based on their direct relevance to FDD or anomaly detection in industrial contexts, emphasizing the practical application and outcomes of the proposed methods or approaches. The detailed criteria for inclusion and exclusion are outlined below, providing a clear framework for selecting and analyzing articles included in this review.

#### 3.1.1. Inclusion Criteria

Original research articles addressing Fault Detection or Diagnosis (FDD) within the context of industrial machinery or processes.Original research articles discussing anomaly detection, provided they pertain to faults in the aforementioned context.

#### 3.1.2. Exclusion Criteria

1.**Initial screening:** articles are first screened based on the following exclusion criteria:**Books and non-article publications:** excluding books, book chapters, and non-scientific articles to focus on original research articles.**Sensor and Monitoring Specific Studies:** Excluding studies specifically focused on computer vision, infrared, thermography, vibration, acoustic, or individual sensor/actuator monitoring, as these often address niche applications within FDD.**Document availability and type issues:** excluding documents not found, not indexed, pre-prints, retracted articles, and those unavailable in full text or not in English to ensure accessibility and academic rigor.**Non-ML focused studies:** studies not dealing with Machine Learning or those focused on data integration and processing, quality deviation detection, and Quantum ML are excluded to maintain focus on ML applications in FDD.**Specific application or context:** studies focused on improving techniques involving time-series, based on Digital Twin or Cyber-Physical Systems (CPS), focused purely on rotary machines or bearings, related to network monitoring and cybersecurity, out of industrial machinery or process context, or dealing with tool condition monitoring are excluded to maintain the scope relevant to general FDD in industrial machinery and processes.**Article type specific:** discussions, general reviews, surveys, studies, and short papers are excluded to prioritize original, in-depth research articles.2.**Secondary screening:** following the initial screening, articles mainly focusing on FDD in rotating machinery components such as motor engines, turbines, pumps, and similar equipment have been excluded.3.**Final selection:** The remaining articles are assessed based on their outcomes. Only those presenting or proposing methods or approaches for FDD or anomaly detection in industrial machinery or processes have been considered for an in-depth review.

Within the extensive dataset of 805 documents, 136 were found to be relevant to Fault Detection and Diagnosis in industrial contexts. Among these, 29 studies were identified as particularly noteworthy for presenting innovative methods that address the complexities and challenges associated with fault detection. Table 3 summarizes each study, encapsulating their objectives, challenges addressed, and techniques employed. The following section discusses the studies based on the purpose and the problem addressed, providing a comprehensive overview of current trends and techniques.

### 3.2. Recovered Studies: Challenges Addressed, and Techniques Employed

To provide a structured analysis of the reviewed studies, we have categorized them based on the types of problems addressed and the techniques employed. The categorization followed an iterative methodology, beginning with an initial proposal refined through multiple discussion cycles and regrouping among the authors. The collaborative effort resulted in the identification of six distinct groups: (1) Data and Process Complexity; (2) Early Anomaly or Novel Fault Detection; (3) New, Improved, Combined, or Hybrid Approaches; (4) Shop-floor Implementation Requirements; (5) Lack of Labeled Data; and (6) Incorporation of Explainability and Interpretability. These groups provide a comprehensive framework for examining the challenges tackled and the methodologies applied in the studies, as detailed in the subsequent sections.

#### 3.2.1. Data and Process Complexity

Ren et al. [13] proposed a methodology based on deep belief networks and multiple models (DBNs-MMs) to detect faults in complex systems. The primary objective was to balance the level of detail in the model to ensure accurate fault detection while keeping the model complexity manageable. The approach relies on an adaptive threshold for fault detection that considers the normal condition of the process. It establishes limits for deviation regarding absolute distance from the normal condition and time of permanence in this state. Although the approach showed promising results, the study highlighted two significant limitations: the relatively small dataset size and the fact that it required several trials, errors, and ad hoc interventions.

Chiu et al. [14] proposed a method to enhance real-time fault detection in Cyber-Physical Systems (CPS) by addressing the computational complexity issue. They utilized Random Forest (RF) and a time-series Deep-Learning model based on Long Short-Term Memory (LSTM) networking to achieve real-time monitoring and enable faster corrective adjustment of machines. As a result, their solution can trigger an alarm with 80% accuracy 3 h before a failure occurs. This allowed shop-floor engineers to adjust its parameters or perform maintenance to mitigate the impact of its shutdown.

Berghout et al. [82] have developed an innovative Machine Learning approach titled “Auto-NAHL: A Neural Network Approach for Condition-Based Maintenance of Complex Industrial Systems” to enhance condition monitoring in industrial processes. Their research focuses on introducing a feature extraction and selection method based on correlation analysis and dimensionality reduction. The group proposed the Automatic Artificial Neural Network with an Augmented Hidden Layer (Auto-NAHL) for classifying health states. This method leverages multiple feature mappings of inputs to provide a more accurate classification. By automating hyperparameter tuning through the Particle Swarm Optimization (PSO) technique, the researchers demonstrated the effectiveness of their approach on a complex industrial plant. They showed that their method outperforms existing health state classification and adaptability methods. This study highlights the potential of their solution in providing real-time health assessments, which can optimize maintenance planning in industrial settings.

#### 3.2.2. Early Anomaly or Novel Fault Detection

Jabbar et al. [83] developed a unique method for identifying conditional anomalies which can be easily deployed in production environments. Their solution is based on Variational Autoencoders (VAEs), which provide interesting scores under the near real-time constraints of the production environment. They evaluated the results with the support of expert process engineers. This close relation with field experts was key to reaching a solution that provides a level of interpretability, and enables users to define trade-offs according to quality an process requirements.

Ketonen et al. [84] developed a novel approach for anomaly detection in injection molding processes. They used probabilistic Deep Learning, which combines Variational Autoencoders (VAEs) and Recurrent Neural Networks (RNNs) to capture both static and dynamic patterns in data. This approach effectively distinguishes between normal and anomalous operation cycles in injection molding machines, making manufacturing processes more reliable and efficient. The method has been evaluated on real-world datasets from injection molding machines, demonstrating its ability to detect various anomalies accurately. The results highlight the potential of integrating probabilistic modeling with Deep Learning techniques in enhancing manufacturing processes.

Lin et al. [85] developed a Machine Learning-based approach for automatically monitoring an ion implanter in semiconductor manufacturing. The authors tested seven Machine Learning algorithms and compared their recipe classification performance using two wafer datasets. The experimental results indicate that the AdaBoost model outperformed the other two models, namely Decision Tree and Random Forest, in detecting anomalies. This study’s findings help improve semiconductor device manufacturing processes by automatically monitoring the ion implanter and detecting anomalies in real time. Overall, this paper contributes significantly to semiconductor manufacturing by demonstrating the effectiveness of Machine Learning-based approaches in automatic monitoring and anomaly detection.

Lorenti et al. [86] developed a novel approach for anomaly detection in manufacturing systems, focusing on Industry 4.0. The study employs a combination of Machine Learning techniques, including clustering and classification, to provide early detection of anomalies in working equipment, thereby ensuring timely alerts to manufacturing supervisors. The critical idea revolves around leveraging the vast amounts of data generated in modern manufacturing environments to enhance the efficiency and reliability of production processes. By integrating these advanced detection methods, the study contributes to minimizing downtime, reducing maintenance costs, and promoting the overall optimization of manufacturing operations in the era of Industry 4.0.

Kim et al. [87] have developed a new approach, based on Machine Learning, for detecting faults in semiconductor etch equipment, both at the process and part levels. The study focused on addressing the limitations of previous methods, especially in detecting equipment abnormalities in mass production settings where severe class imbalances are common. By utilizing Machine Learning techniques, the authors aimed to improve the accuracy of predicting output data in advance. Their approach emphasized the importance of detecting process drift anomalies accurately and effectively classifying equipment part malfunctions to prevent wafer misprocessing. Through their methodology, they demonstrated the potential of Machine Learning in increasing the likelihood of successful predictions, highlighting its significance in semiconductor manufacturing.

#### 3.2.3. New, Improved, Combined, or Hybrid Approaches

Theljani et al. [88] have introduced a new technique for identifying faults in industrial systems in real time by utilizing the concept of density variation in clustering algorithms. The authors recognize the challenges posed by traditional clustering methods, such as k-means and spectral clustering, which often fail in real-world scenarios due to their reliance on distance measurements. To overcome this challenge, they introduced a method focusing on the variation of inner density (derivative) rather than the density itself. This innovative approach effectively mitigates the curse of density heterogeneity commonly encountered in density-based algorithms. The proposed solution effectively detects anomalies as they occur, giving it a significant advantage for real-time industrial applications. However, the algorithm cannot currently localize faults, which the authors plan to address in future works.

Ruan et al. [89] have developed an advanced Deep Learning-based approach called CURNet for predicting faults in process industries. They have used real-time sensory data collected from a chemical plant in their study. Additionally, they have proposed a novel learning algorithm called RGD for CURNet. The study results have demonstrated that CURNet has outperformed the existing time-series fault prediction models when it comes to fault prediction recall and fault type classification accuracy. This research has highlighted the effectiveness of Deep Learning-based anomaly detection in industrial processes and its potential for improving the fault prediction in cyber-physical systems.

Chien et al. [90] developed a Convolutional Neural Network (CNN) model for detecting and diagnosing faults in semiconductor manufacturing processes. They proposed a CNN architecture that effectively handles time-series sensor data commonly encountered in semiconductor manufacturing. The model transforms time-series data into a 2-dimensional array, similar to an image structure. The SVID and timestamp are represented on the x and y-axis, respectively, and the sensor measurement values are treated as pixel values. The CNN model extracts fault features along the time axis to represent process faults. The authors also highlighted the importance of feature extraction for the validity of the classification model. They suggested using pre-trained CNN models such as AlexNet and VGG16/19 for image recognition and classification tasks in semiconductor manufacturing. The authors emphasized the need for further research using image recognition technology for time series fault detection and classification (FDC) data.

Selvanathan et al. [91] have developed a new method for detecting faults in multi-component industrial systems at an early stage. They used Cascaded CNN-LSTM (CC-LSTM) models to perform this task. The key idea behind their approach is to leverage the strengths of Convolutional Neural Networks (CNNs) and Long Short-Term Memory networks (LSTMs) in a cascaded manner. CNNs can extract complex feature representations from the input windowed data and capture spatial information. LSTMs, on the other hand, process these features to capture temporal dependencies, making them suitable for time-series classification. The proposed CC-LSTM models classify sensor data at each time step into ‘normal behavior’ or ‘abnormal behavior’, which are then used to determine the onset of abnormal operation. The study demonstrated the efficacy of the CC-LSTM approach in detecting early signatures of faults and highlighted its superior performance compared to other Deep Learning techniques.

Askari et al. [92] devised a novel unsupervised learning approach for Real-Time Fault Detection and Diagnosis (RT-FDD) in industrial systems. Their approach involves using clustering techniques, particularly the k-means algorithm, to analyze and convert raw monitoring data into valuable insights for Condition Based Monitoring (CBM) applications. By employing this strategy, they aimed to enhance the efficiency of process monitoring and fault diagnosis in modern automation systems. The study highlights the growing importance of Prognostics and Health Management (PHM) in implementing predictive maintenance within complex industrial frameworks. It emphasizes that timely and accurate fault detection can reduce damage, minimize maintenance costs, and increase productivity.

Cohen et al. [93] proposed a new method for detecting faults in industrial systems in real time. They combined Machine Learning techniques with Petri nets, a tool to model and analyze the system’s behavior. Their approach combines physics-based and data-driven methods, leveraging both strengths and mitigating their limitations when used separately. Their solution showed promising results in detecting and diagnosing faults in real-world industrial scenarios. The authors highlight the potential of Petri nets in enhancing system reliability and reducing downtime in industrial settings.

#### 3.2.4. Shop-Floor Implementation Requirements

Westbrink et al. [3] have proposed a novel approach to Machine Learning called “Machine Proximity Machine Learning”. They achieved this by using an Industrial PC (IPC) based on the Raspberry PI platform installed close to the machine’s controller and connected directly to it. This approach offers a low-cost, secure, and easy way to apply ML in industrial applications without needing expensive PCs or sharing data with cloud platforms.

Leite et al. [49] have developed a novel technique for detecting and diagnosing faults in discrete manufacturing machinery in real time using Automated Machine Learning (AutoML). Their methodology is designed to eliminate the need for Machine Learning experts during implementation, making it highly accessible for industries. The central idea of their research is to combine continuous variables and discrete events to improve the accuracy and efficiency of fault diagnosis. Their approach showed an exceptional ability to diagnose faults right from the initial deployment and demonstrated potential for further improvement with each new detected fault. These results highlight the effectiveness of their method and emphasize its potential benefits for industries that aim to utilize Machine Learning for fault detection without the need for specialized expertise on the shop floor.

Wang et al. [94] have designed a Machine Learning-based system aimed at detecting entanglement issues in older dyeing machines, which is a common problem in dyeing and finishing factories. The study aimed to address the challenge of data imbalance inherent in the dyeing process by utilizing the capabilities of Extreme Gradient Boosting (XGBoost) and Random Forest (RF) models. The proposed system acts as an early warning mechanism, improving the dyeing quality by predicting potential entanglements. The XGBoost model showcased significant improvements, with an F1 score as high as 94%, while the Random Forest method demonstrated commendable classification performance with an F1 score of 92%. This innovative approach offers dyeing and finishing factories a robust solution to mitigate quality issues and optimize their processes.

#### 3.2.5. Lack of Labeled Data

Furukawa et al. [15] developed a method for early fault detection on sensor systems by combining ChangeFinder (CF) and SVM. Their method uses the change score generated by the ChangeFinder (CF) as new features at the SVM for classifying normal and anomalous conditions. The authors show that the ChangeFinder-SVM (CF-SVM) improved detection speed and accuracy compared to the original SVM. This is particularly novel as it addresses the challenge of early fault detection in industries that typically lack labeled data for Machine Learning. Their work provides a benchmark for comparing future methods, showcasing the effectiveness of combining statistical change detection with Machine Learning for industrial applications.

Quatrini et al. [95] incorporated the cyclic behavior of processes into the dataset by introducing a phase labeling step and categorizing the anomalies in “expected”, “warning”, and “critical”. The study found that after introducing the phase labeling step, anomaly detection successfully detected anomalies with precision and recall rates exceeding 99%. However, without this step, the anomaly detection system failed to detect warning and critical situations in the production phases.

Soller et al. [96] developed an anomaly detection system tailored for industrial machinery in production environments. The core idea behind their research is to continuously monitor machine data to identify any accumulations of anomalies that could potentially precede or lead to machine defects or faulty behavior in the produced products. By leveraging data collected over several months from industries, the team has successfully designed a system that predicts errors and significantly reduces the need for manual inspections. This innovative approach has shown great promise in enhancing error identification, leading to more sustainable and efficient production processes. The system’s ability to act as a reliable indicator for any machine malfunction or product defects showcases its potential to revolutionize fault detection in industrial settings.

Kolokas et al. [97] have developed a novel method for predicting faults in industrial machinery. They used unsupervised anomaly detection using historical process data and then correlated it to upcoming equipment faults using binary classification. As a result, they can enable real-time predictive maintenance. The approach has been tested on aluminum and plastic production industries, and the model achieved a significant Matthews Correlation Coefficient of up to 0.73. This research demonstrates that the algorithm broadly applies across multiple sectors, promising more comprehensive implementation and predictive accuracy in different industries.

Calabrese et al. [98] have developed a methodology that can detect and diagnose faults in industrial machinery in real time. They have used a combination of clustering and classification techniques to address the challenge of limited availability of fault data in industrial settings. The key idea behind their approach is to leverage the strengths of both unsupervised and supervised learning. The methodology can identify known and novel faults during machinery operation by initially clustering data to identify potential anomalies and then employing classification models. The proposed solution has been successfully applied to the sealing sub-system of a packaging machine, demonstrating its capability for accurate and efficient fault detection with minimal training data. Therefore, it offers a promising tool for improving machinery reliability and safety in industrial environments.

#### 3.2.6. Incorporate Explainability and Interpretability

Singhal et al. [99] proposed a novel approach for fault diagnosis in industrial systems, leveraging the capabilities of Long Short-Term Memory (LSTM) networks. The study emphasizes the LSTM’s proficiency in capturing long-term temporal dependencies in time-sequence data, making it well-suited for fault classification. The authors’ proposed methodology not only detects and identifies faults, but also aims to identify their root causes. The approach demonstrates a diagnosis accuracy of approximately 71%, showing promising results for the industrial test bed examined; the paper acknowledges the need for further investigation regarding its applicability to other systems, especially those with multiple components and significant process lags. The study underscores the potential of LSTM-based models in enhancing industrial systems’ efficiency, productivity, and quality by enabling timely fault detection and resolution.

Koujok and his team [11] have developed an approach to build a decision support tool (DST) that assists operators in large-scale processes to diagnose faults and make correct decisions. They combined supervised and semi-supervised techniques to detect and diagnose faults, concluding that supervised methods are inefficient in providing meaningful explanations and detecting unknown faults. To overcome these limitations, the team has explored integrating semi-supervised methods into the process. In a subsequent article [12], they propose improving the previous work using a multi-agent-based methodology. These studies have resulted in a method that can detect new faults, diagnose known faults, and provide valuable information to maintenance professionals.

Castellanos et al. [100] propose a particular assembly of Classification and Regression Trees (CART). Their algorithm has two steps: the first detects fault states, and if it is faulty, the second determines the type of fault. Their results show that the proposed solution improved the balance in classification and the interpretability of the diagnosis. Their innovative method improves the balance in classification performance, providing clear insights into the model’s decision-making process, which is crucial for interpretability in fault diagnosis within complex industrial systems.

Bhakti et al. [101] proposed a method for identifying the variables responsible for a Deep Neural Network (DNN), classifying an input as a fault. The method is based on the Shapley value framework and utilizes integrated gradients for implementation. The approach estimates the marginal contribution of each variable to the DNN, averaged over the path from the baseline (normal state) to the current sample. This variable attribution method provides sample-specific explanations of the DNN’s prediction, making it easier to understand and interpret the DNN’s results.

Zhang et al. [69] have developed a new approach to classify time series data generated by industrial manufacturing processes, particularly for the continuous casting process. The main idea behind their approach is to extract representative subseries from the time series data using an optimal shapelet combination to improve classification accuracy. This method is designed to address the challenges of real-time classification in a big data environment while prioritizing the importance of interpretability. As a result, their approach can efficiently process vast amounts of high-resolution data and provide clear and intuitive classification results. This method has significant potential in guiding production and management decisions in industrial settings by enabling fast and accurate time series data classification.

The researchers Ameli et al. [102] have developed a new method called “Explainable Unsupervised Multi-Sensor Industrial Anomaly Detection and Categorization”. This method is focused on detecting and categorizing anomalies in data collected from multiple sensors in industrial settings. The study has been conducted as part of the SPAICER project, and is a component of the Smart Resilience Services. The approach uses Explainable AI to locate anomalous sensors and provides interpretable insights into potential issues. The results of the research showcase aggregated anomaly scores from eight runs of glass production data. This work aims to improve the resilience of industrial processes by providing timely and interpretable insights into potential problems, ensuring that necessary actions can be taken promptly to avoid loss of product quality or further complications.

**Table 3 sensors-25-00060-t003:** Summary of selected studies categorized by addressed challenges.

Purpose	Study	Year	Goals/Objectives	Problems Addressed	Techniques Used
Deal with	[13]	2018	Methodology for complex system fault detection	Model balance and complexity	Deep Belief Networks, Multiple Models
Process	[14]	2020	Real-time fault detection improvement	Computational complexity in CPS	Random Forest, LSTM
Complexity	[82]	2021	Condition-based maintenance enhancement	Health state classification	Auto-NAHL, PSO technique
Early Anomaly	[83]	2019	Conditional anomaly detection	Real-time constraints in production	Variational Autoencoders
or Novel Fault	[84]	2021	Anomaly detection in injection molding	Static and dynamic pattern capture	Variational Autoencoders, RNNs
Detection	[86]	2022	Anomaly detection in Industry 4.0	Equipment anomaly detection	Machine Learning techniques
	[85]	2022	Automatic monitoring of ion implanter	Anomaly detection in semiconductor manufacturing	Machine Learning algorithms
	[87]	2022	Process-level fault detection	Equipment abnormalities detection	Machine Learning techniques
New, Improved	[15]	2020	Early fault detection in sensors	Detection speed and accuracy	ChangeFinder, SVM
or Combined	[88]	2021	Real-time fault detection	Density variation in clustering	Density-based clustering
Techniques	[89]	2022	End-to-end fault prediction	Fault prediction in process industries	CURNet, RGD learning algorithm
	[90]	2022	Fault detection and diagnosis	Time-series data handling in manufacturing	Convolutional Neural Network
	[91]	2022	Early fault detection in multi-component systems	Fault detection in complex systems	Cascaded CNN-LSTM models
	[92]	2023	RT-FDD in industrial systems	Efficiency in process monitoring	Clustering techniques, k-means
	[93]	2022	Real-time fault detection	Integration of physics-based and data-driven methods	Machine Learning, Petri nets
Deal With	[3]	2018	ML application in industrial settings	Secure and cost-effective ML implementation	Industrial PC, Raspberry PI
Shop-floor	[49]	2022	RT-FDD in Discrete Manufacturing Machines	ML based fault diagnosis accessibility	Automated Machine Learning
Requirements	[94]	2022	Entanglement detection in dyeing machines	Data imbalance in dyeing process	Extreme Gradient Boosting, Random Forest
Address	[95]	2020	Improved anomaly detection	Phase labeling in datasets	Cyclic behavior incorporation
the Lack of	[96]	2021	Anomaly detection in industrial machinery	Error identification	Machine data monitoring
Labeled Data	[97]	2020	Fault prediction in industrial equipment	Correlating anomalies to faults	Unsupervised anomaly detection, binary classification
	[98]	2021	RT-FDD in industrial machinery	Limited fault data availability	Clustering, classification techniques
Incorporate	[99]	2019	Fault diagnosis with root cause identification	Temporal dependencies in data	Long Short-Term Memory networks
Explainability	[11]	2019	Decision support tool development	Fault diagnosis in large-scale processes	Supervised and semi-supervised techniques,
and/or	[12]	2021			multi-agent methodology
Interpretability	[100]	2020	Improved fault state detection and classification	Balance in classification	CART
	[101]	2022	Variable identification leading to faults	Interpretation of DNN predictions	Shapley value framework, integrated gradients
	[69]	2021	Time series classification in manufacturing	Real-time classification in big data	Optimal shapelet combination
	[102]	2022	Industrial anomaly detection	Interpretation of multi-sensor data	Explainable AI, anomaly detection

## 4. Findings, Challenges, and Prospects for Future Research

### 4.1. RT-FDD in the Industrial Domain

During the comprehensive literature review, several key findings emerged regarding the application and assessment of Machine Learning in industrial settings. Out of the 29 studies analyzed, 17 used actual industrial data for testing or evaluation, indicating a clear trend toward using authentic data sources in current research [11,12,14,69,83,84,85,86,88,89,90,92,94,98,100,101,102]. A temporal analysis of these studies revealed a growing trend. While only 15 studies were published up to 2021, there was a significant increase in the subsequent 1.5 years, with 14 new studies emerging in 2022 and 2023. This suggests a rapid acceleration and heightened interest in this field recently.

The semiconductor manufacturer sector alone had four dedicated studies [85,87,90,93], while the remaining research covered a diverse range of sectors such as general continuous processes, injection molding machines, work cells, float glass production, tank systems, assembly machines, printers, bulk good laboratory plants, casting processes, compressors, cryogenic propellant loading systems, Discrete Manufacturing Machines, dyeing and finishing factories, and hydraulic systems.

Regarding the industrial domain, only two studies provided explicit information about their use of Machine Learning in an industrial setting [14,94], evidencing that most studies have a clear gap in the real-world impact of implementing Machine Learning methodologies. Moreover, only 7 out of the 29 studies used benchmark datasets essential for reliable comparative evaluations [11,12,13,69,82,99,101]. Furthermore, none of the studies that used other datasets made them publicly accessible, which poses challenges for reproducing their results and conducting future research.

Furthermore, a glaring gap has been observed in comparative evaluations, except for studies published sequentially by the same research group [11,12], no study engaged in a comprehensive comparison with other contemporary approaches. Most studies only compare their results with established Machine Learning algorithms or techniques without comprehensively comparing them with other contemporary approaches. As a result, recent methodological advancements have yet to be compared. This gap in research needs to be addressed to advance the field and improve outcomes.

It is worth noting that most studies on Real-Time Fault Detection and Diagnosis (RT-FDD) have relied on time series data. However, the need for more information on the feature extraction processes used in these studies suggests there is potential for improvement in future research.

### 4.2. Literature Review Research Questions

#### 4.2.1. State-of-the-Art in RT-FDD (LRRQ1)

It is difficult to determine the state of the art in Fault Detection and Diagnosis (FDD) as different studies use different datasets in their experiments. Only two studies use the Tennessee-Eastman Dataset (TED) [11,12,101], while another four studies used different benchmark datasets [13,69,82,99]. Moreover, TED is suitable for studying FDD on continuous processes, not discrete systems. Therefore, comparing the performance of a new approach to others requires the researcher to implement the other approaches. Each study deals with unique aspects not present in others, making it challenging to present a comprehensive overview of the proposed approach compared to others.

#### 4.2.2. Primary Research Topics in RT-FDD (LRRQ2)

Numerous research efforts have been undertaken in Real-Time Fault Detection and Diagnosis (RT-FDD) to enhance the efficacy and reliability of monitoring systems across various industrial domains. These endeavors span various methodologies and application areas, reflecting the diverse challenges encountered in this field.

A fundamental approach to RT-FDD involves the development of Decision Support Tools (DST). Koujok and their team have exemplified this approach by creating a DST that helps operators diagnose faults in large-scale processes. Their work combines supervised and semi-supervised techniques to address the limitations of supervised methods by providing meaningful explanations and detecting previously unknown faults [11,12]. This innovation is a significant advancement in operational diagnostics.

In parallel with DST advancements, Deep Learning has emerged as an essential strategy for managing the complexities of fault detection in various systems. Ren et al. applied deep belief networks and multiple models to improve fault detection capabilities in complex systems [13]. Meanwhile, Chiu et al. focused on cyber-physical systems (CPS). They used Random Forest and LSTM networks to tackle the computational challenges inherent in CPS fault detection [14]. These studies highlight the potential of Deep Learning techniques in overcoming the intricate challenges presented by complex industrial environments.

Innovative methodologies for anomaly detection are being introduced in Real-Time Fault Detection and Diagnosis (RT-FDD). For example, Jabbar et al. recently used Variational Autoencoders (VAEs) to detect conditional anomalies within production settings [83], while Furukawa et al. employed ChangeFinder and SVM for early fault detection in sensor systems [15]. Such techniques show the evolution of anomaly detection methods, which are crucial for preemptive fault identification.

Exploring various Machine Learning strategies further enriches the research landscape in RT-FDD. Castellanos et al. have introduced a unique assembly of Classification and Regression Trees (CART) for Fault Detection and Diagnosis [100], while Westbrink et al. have presented an approach that leverages an Industrial PC based on the Raspberry PI platform, demonstrating the feasibility of applying Machine Learning in industrial settings in a cost-effective manner [3]. Additionally, the integration of cyclic behavior into datasets for improved anomaly detection by Quatrini et al. [95], and the predictive fault detection methods developed by Kolokas et al. [97] highlight the diverse avenues through which Machine Learning is being applied to enhance RT-FDD.

Several researchers have significantly contributed to enhancing the explainability and predictability of fault detection systems. Bhakte et al. have utilized the Shapley value framework to identify critical variables in Deep Neural Network (DNN) classifications [101], while Ruan et al. have developed an end-to-end Deep Learning approach for fault prediction in process industries [89]. These advancements indicate the growing emphasis on understanding and predicting faults more precisely.

Innovative approaches have also been proposed to address the specific challenges of semiconductor manufacturing, such as the CNN model for Fault Detection and Diagnosis proposed by Chien et al. [90]. The Machine Learning-based automatic monitoring of ion implanters by Lin et al. [85]. These contributions demonstrate the sector-specific applications of RT-FDD technologies.

Moreover, the interdisciplinary nature of Real-Time Fault Detection and Diagnosis (RT-FDD) research is evidenced by the integration of Machine Learning techniques with Petri nets by Cohen et al. [93] and the combination of clustering and classification techniques by Calabrese et al. [98]. Ameli et al. [102] have developed Explainable AI methods for industrial anomaly detection. In contrast, Leite et al. [49] have utilized Automated Machine Learning (AutoML). These innovative strategies exemplify the varied approaches employed to meet Real-Time Fault Detection and Diagnosis challenges in an increasingly complex and automated industrial landscape.

The exploration of RT-FDD systems in the field of Deep Learning and Artificial Intelligence (AI) showed a transformative trajectory in predictive maintenance and operational optimization across various industrial applications. In the study by Russul H. Hadi et al. [103], the advent of AutoML has been harnessed to significantly enhance fault classification in industrial IoT settings, emphasizing its efficacy in ball bearing fault identification with a notable reduction in manual parameter tuning and computational resources. Similarly, the work by Xueteng Sun et al. [104] introduces a hybrid Shallow and Deep Convolutional Neural Network (SDCNN) model, which integrates Deep Learning for chiller fault diagnosis, demonstrating superior performance in detecting and diagnosing chiller anomalies. Gaowei Xu et al. [105] methodology leverages transfer Convolutional Neural Networks, presenting a novel approach for online fault diagnosis that efficiently addresses the complexities of industrial equipment monitoring. Huifang Li et al. [106] further delved into the realm of Deep Learning by applying Binarized Deep Neural Networks combined with Random Forests, offering an efficient and scalable solution for large-scale rotating machinery fault diagnosis. Lastly, the work by Saúl Langarica et al. [107] exemplifies the integration of domain knowledge with probabilistic AI to refine Fault Detection and Diagnosis in industrial motors, illustrating the potential of intelligent systems in enhancing the reliability and efficiency of industrial operations. Fangzhou Guo et al. [108] push the envelope by implementing a domain knowledge-incorporated and probabilistic AI-enhanced method for diagnosing faults in electric bus air conditioning systems, showcasing its robustness even under the dynamic operating conditions of electric buses.

Overall, RT-FDD research encompasses a wide array of methodologies, from Deep Learning and Machine Learning techniques to incorporating innovative algorithms and frameworks. These collective efforts are directed toward improving fault detection systems’ precision, efficacy, and clarity, guaranteeing their flexibility in response to the changing demands of diverse industrial sectors.

#### 4.2.3. ML Techniques Employed for RT-FDD (LRRQ3)

Regarding the ML techniques, in general, studies use unsupervised or semi-supervised techniques such as One-Class SVM [15,96], Isolation Forest [86,96] or VAE [83,96] to perform unsupervised anomaly detection tasks, or PCA [3,12] or LSTM [14,89,91,99] for semisupervised anomaly detection with a ad hoc trade-off definition. Moreover, The diagnosis task is commonly held by classifiers, such as SVM [15], RF [12,14,94,95], or CART [100], AdaBoost [85], Shapelet with Genetic Algorithms [69], and Gradient Boosting [87,94].

Some studies have incorporated different approaches to improve the detection speed or reduce the system’s complexity. For instance, one study used the ChangeFinder [15] algorithm to increase SVM’s detection speed, while another one used M-RBC [12] for extracting the features that explained the detected novel anomaly and support operators analysis. Moreover, one other study based the approach on DBNs-MMs [13] to reduce the system’s complexity. Additionally, some studies have applied new paradigms such as Convolutional Neural Networks [90,91] and Cyber-Physical Systems to address Real-Time Fault Detection and Diagnosis tasks [14,89,109].

#### 4.2.4. Challenges to Implement ML Based RT-FDD in Industry (LRRQ4)

The implementation of Real-Time Fault Detection and Diagnosis (RT-FDD) technologies on the shop floor is limited due to various challenges across different industries. These challenges range from the technical complexities of Machine Learning models to operational and resource-related constraints, highlighting the difficulty of integrating advanced diagnostic tools into existing industrial frameworks.

One significant barrier to adopting Machine Learning models is their inherent need for interpretability. Despite their potential for accurate predictions, these models often need to provide clear insights into the logic underpinning their decisions. This lack of transparency, commonly called the “black-box” nature of Machine Learning, erodes trust among plant personnel. Employees are reluctant to depend on systems whose reasoning processes need to be more transparent to them [12,69,87,90,101]. This is a widespread issue since Machine Learning models’ complexity and extensive parameterization obfuscate their decision-making processes, making validation and verification efforts more complicated. Such validation is indispensable in sectors where accountability and safety are paramount, making the lack of transparency a critical obstacle [101].

The effectiveness of Machine Learning models depends on the availability and quality of the data used to train them. However, many industries need help to obtain comprehensive and reliable datasets. Only complete, consistent, or biased data can ensure the performance and reliability of these models, making it difficult for organizations to adopt them [90,93,98,101]. Additionally, the imbalance of operational data, where normal operations are much more frequent than faults, also challenges data-driven approaches. This can make it challenging for models to accurately identify potential failures, hindering the development of intelligent manufacturing solutions [87,89,90,94].

Integrating RT-FDD technologies into existing infrastructures can also be challenging. Incorporating Machine Learning models into current industrial ecosystems requires significant changes to infrastructure, data collection mechanisms, and day-to-day operations. This complexity and resource intensity can act as a deterrent for organizations considering the adoption of Machine Learning solutions [101].

Implementing Machine Learning in industrial settings is a complex process that requires specialized expertise and significant resources. These systems’ development, deployment, and management call for skilled data scientists and Machine Learning professionals, which poses a significant challenge. Additionally, the financial costs associated with procuring and maintaining the requisite computational infrastructure can be prohibitive for many industries [3,49,82,99,101].

The journey toward widespread adoption of RT-FDD technologies in industrial environments is filled with technical and data-related issues, operational constraints, and resource limitations. Overcoming these challenges is essential to unleash the full potential of Machine Learning in enhancing Fault Detection and Diagnosis processes, which will contribute to the evolution of intelligent manufacturing practices.

#### 4.2.5. Potential Benefits Implementing ML Based RT-FDD in Industry (LRRQ5)

Integrating Machine Learning (ML)-based Real-Time Fault Detection and Diagnosis (RT-FDD) systems in industrial settings offers significant advantages that can transform maintenance strategies and enhance operational efficiencies. One primary benefit of using ML in RT-FDD is the substantial improvement in fault prediction accuracy. Machine Learning techniques excel at analyzing large volumes of sensory data in real time, identifying patterns and anomalies that could indicate potential faults. This capability enables early detection and prediction of system malfunctions, allowing industries to implement proactive maintenance strategies that can significantly minimize downtime and improve operational continuity [11,49,89,92,99,100].

Furthermore, ML models are notable for their capability to improve and enhance their predictive accuracy over time. By analyzing historical data, these models refine their understanding of system behaviors, thus improving their ability to accurately forecast potential faults. This continuous learning process reduces the occurrence of false alarms and strengthens the overall reliability of industrial systems [85,87,89,90]. Another significant advantage of ML models is their adaptability to complex industrial systems. Modern industrial processes are complex, often involving intricate systems with numerous sensors and variables. ML algorithms can effectively manage this complexity by navigating the nonlinear relationships between different variables, providing reliable Fault Detection and Diagnosis in multifaceted environments [82,89].

Real-time monitoring is a crucial feature of ML-based RT-FDD systems. It enables continuous surveillance of industrial processes, allowing for instant fault identification and timely interventions to prevent further damage or operational disruptions. This capability enhances system responsiveness and helps maintain high levels of process integrity [49,82,85,89]. From a financial perspective, ML-based systems that detect faults can result in significant cost savings. By enabling proactive maintenance, these systems help avoid costly breakdowns and unplanned downtime, allowing for efficient scheduling of repairs or replacements. This proactive approach optimizes resource utilization and minimizes maintenance expenditures, leading to improved financial outcomes for industrial operations [89,93,100].

Lastly, the scalability of Machine Learning models provides a crucial advantage for industrial applications. As industrial processes and data generation expand, ML models can be trained on increasingly large datasets. This enables them to handle growing data volumes without sacrificing performance. This scalability ensures that ML-based RT-FDD systems can adjust to the changing data requirements of industrial environments, maintaining their effectiveness and reliability as operational scales expand [49,82,89].

#### 4.2.6. RT-FDD in Continuous Process and Discrete Manufacturing (LRRQ6)

Until 2021, most FDD studies focused on continuous processes with stable conditions, making anomaly detection easier. Discrete Manufacturing Machines (DMMs), with their cyclic and sequential behaviors, introduce dynamic challenges often overlooked, as earlier studies relied on aggregated continuous variables.

Since 2022, research on DMMs has grown significantly, though only four studies explicitly addressed their cyclic and sequential behaviors [49,92,93,99]. These works emphasize combining continuous and discrete data to effectively capture fault signatures unique to DMMs.

Advancing RT-FDD for DMMs is critical to enhancing fault detection accuracy, enabling predictive maintenance, and boosting productivity in modern manufacturing systems.

### 4.3. Prospects for Future Research (LRRQ7)

The literature review on integrating and evaluating Machine Learning in industrial contexts revealed several areas for further exploration and research. The following are some of the prominent research opportunities identified:1.**Standardized datasets:** The absence of benchmark datasets in over half of the studies and the lack of publicly accessible datasets underscore the need for standardized datasets in the domain. Creating and promoting standardized datasets can foster reproducibility and facilitate comparative evaluations, propelling the field forward.2.**Unbalanced and balanced datasets:** Databases that contain faults are often unbalanced due to the rarity of fault occurrences, challenges in detecting faults, and biases in data collection processes. This class imbalance challenges Machine Learning algorithms as non-fault instances outnumber fault instances. As a result, Machine Learning techniques such as resampling and ensemble methods must be used to ensure effective fault detection. One potential research opportunity is the ability to simulate machine behavior and create databases by defining behavior models that capture essential parameters and relationships that affect machine operations. By exploring machines with different dynamics and accounting for variations in behavior over time, it is possible to generate diverse and realistic synthetic data that represents various operating conditions, including anomalies and faults.3.**Feature extraction in time series data:** Researchers are exploring advanced techniques such as Recurrent Neural Networks (RNNs) and Convolutional Neural Networks (CNNs) for improved analysis and methods for incorporating uncertainty estimates and handling multivariate data. Additionally, there is growing interest in interpretable models and techniques for anomaly detection in streaming data. Since fault detection often requires the analysis of temporal series due to the dynamic behavior of systems, it is expected that possible research can delve deeper into innovative feature extraction techniques tailored or implemented to industrial contexts, further advancing fault detection methodologies.4.**Comparative evaluations:** As commented in the discussion, it is difficult to determine the state-of-the-art in FDD. Each study deals with unique aspects not present in others, revealing a glaring absence of comprehensive comparisons with contemporary approaches. Future studies can compare novel methodologies with recent advancements to offer a more transparent state-of-the-art picture.5.**Reducing dependency on specialized expertise:** A key observation is that most ML methodologies require specialized expertise for effective industrial implementation. Research aimed at simplifying these methodologies or developing user-friendly and pedagogic interfaces can make ML-based RT-FDD more accessible to industries.6.**Enhancing interpretability and trust:** Research into Explainable AI (XAI) on enhancing AI models’ interpretability, transparency, and trustworthiness. This involves developing Machine Learning models that are easier to understand, as well as post hoc interpretability techniques that explain complex black-box models. Additionally, efforts focus on interactive visualizations and dashboards for exploring model behavior, counterfactual explanations for understanding decision-making processes, and methods for certifying AI system safety and fairness. Key research areas include incorporating human-centered design principles and addressing ethical and societal implications. In the future, research in XAI should focus on techniques that improve the transparency of ML models in fault and diagnosis detection in the industry.7.**Real-world implementation outcomes:** While many studies utilize real industrial data, there is a noticeable gap in documenting the tangible outcomes or broader impacts of implementing Machine Learning methodologies in real-world industrial settings. Research that bridges this gap can provide invaluable insights for industries contemplating the adoption of ML-based RT-FDD.8.**Addressing data challenges:** With data quality, availability, and imbalance being significant concerns, research that offers solutions to these challenges can be pivotal. Techniques for data augmentation, handling imbalanced datasets, and ensuring data quality can be potential areas of exploration.9.**Enhancing real-time capabilities and integration:** Given the emphasis on Real-Time Fault Detection and Diagnosis, research focusing on optimizing ML models for real-time performance, reducing computational overheads, and ensuring timely fault detection can be crucial. Research that offers seamless integration solutions or frameworks can be of immense value to industries.10.**Exploring novel architectures and techniques for fault detection:** This review did not comprehensively explore advanced methodologies such as causal reasoning, qualitative modeling, the KAN-HyperMP model [110], Graph Neural Networks (GNNs) [111], other specialized Machine Learning and AI techniques, or hybrid approaches, tailored for FDD. Future research could delve into these and other innovative architectures since these advancements align with Industry 4.0’s vision of leveraging sophisticated technologies to optimize industrial processes.

## 5. Discussion and Insights

This literature review highlights both the challenges and opportunities within Real-Time Fault Detection and Diagnosis (RT-FDD) in industrial settings. While the potential for transforming industrial operations is evident, technical, data-related, and integration challenges remain significant barriers to adoption. This section elaborates on these challenges and outlines actionable future research directions to bridge gaps in both theory and practice.

The technical complexity of RT-FDD systems requires expertise often unavailable on the shop floor. Simplifying implementation through Automated Machine Learning (AutoML) can address this issue by automating key processes such as model selection and feature engineering. However, adapting AutoML to diverse industrial settings and ensuring computational efficiency for real-time applications remain open challenges. Overcoming these barriers is crucial for making RT-FDD solutions more accessible and fostering broader adoption across industries, supporting the goals of Industry 4.0 by enhancing automation and operational intelligence.

Data-related challenges, including scarcity, imbalance, and isolation, significantly impede the implementation of RT-FDD systems. The infrequency of fault events in industrial environments necessitates innovative strategies, such as simulation-based synthetic data generation, to create realistic and balanced datasets that can enhance model robustness. However, ensuring the accurate representation of diverse fault scenarios and integrating synthetic data with real-world datasets remain areas requiring further research. Notably, no study has provided comprehensive information on critical aspects such as fault composition, structure, and fault data distribution, which limits the ability to fully understand and address these challenges in real-world applications. Simultaneously, resistance to data sharing, fragmented data standards, and proprietary systems contribute to data isolation, restricting interoperability and scalability. Addressing these challenges aligns with the principles of Industry 4.0, which emphasizes data integration, connectivity, and collaboration to unlock the full potential of smart manufacturing.

The absence of standardized datasets and unified evaluation methodologies further compounds these issues, limiting the comparability of studies and impeding the establishment of a clear state of the art. Developing open-access datasets and shared benchmarks would not only support systematic evaluations, but also promote collaboration between academia and industry. By providing a common foundation for innovation, such standardization initiatives could accelerate progress in RT-FDD research and applications, ultimately driving the development of robust, scalable, and widely applicable solutions.

Integration with legacy systems and addressing cybersecurity concerns are pivotal challenges for implementing RT-FDD technologies. Edge computing offers a practical solution by enabling localized data processing, reducing latency, and enhancing security. Conducting pilot studies to explore scalability and interoperability in diverse industrial contexts can help streamline integration pathways and drive practical implementation.

The interpretability of Machine Learning models is another critical barrier. The “black-box” nature of many algorithms reduces trust among users, particularly in high-stakes industrial applications. Developing Explainable AI (XAI) tailored for RT-FDD, with tools such as visualizations and variable attribution techniques, can enhance transparency, build trust, and improve the reliability and usability of these systems.

Emerging technologies like digital twins and the Industrial Internet of Things (IIoT) offer new possibilities for RT-FDD by enabling simulation, optimization, and predictive maintenance. These technologies are foundational components of Industry 4.0, providing the connectivity and intelligence needed to create adaptive, self-optimizing systems. However, real-time synchronization and the computational demands of integrating these technologies require further research and development to ensure their practical applicability in industrial environments.

This review offers significant contributions compared to prior research, as highlighted in Table 4. Unlike earlier works that often focus on niche areas, such as rotary machines or specific industrial contexts like semiconductor manufacturing [51,53,54], this study provides a broader perspective on RT-FDD, encompassing both continuous and discrete manufacturing systems. It addresses the absence of standardization in datasets, a recurring limitation in past reviews, by emphasizing the urgent need for publicly accessible, benchmarked datasets to ensure reproducibility and comparative evaluations. Additionally, while previous studies have largely discussed established techniques like PCA and SVM, this review incorporates contemporary advancements such as Explainable AI (XAI), Automated Machine Learning (AutoML), and hybrid methodologies. By categorizing studies into six thematic groups, this review facilitates a clearer understanding of RT-FDD challenges and opportunities, creating a structured framework for addressing gaps and fostering synergies across research areas. These contributions position this review as a comprehensive resource, bridging theoretical advancements and practical applications in the context of Industry 4.0.

For instance, grouping studies addressing Explainability and Interpretability (Group 6) highlights a growing recognition of the importance of transparent algorithms in industrial adoption. Similarly, studies in Group 1 emphasize the increasing complexity of data and processes, reinforcing the need for scalable, robust solutions. The categorization also uncovers gaps, such as the underexplored synergy between Hybrid Approaches (Group 3) and Shop-floor Implementation Requirements (Group 4), which is critical for bridging the divide between research innovation and practical deployment.

Future work addressing these challenges promises to strengthen both the theoretical foundations and practical applications of RT-FDD. Simplified methodologies, enhanced data handling strategies, and improved transparency can facilitate widespread adoption, contributing to more efficient, reliable, and sustainable industrial operations. These efforts will shape the trajectory of RT-FDD, balancing academic progress with industrial needs, and advancing the transformative vision of Industry 4.0.

## 6. Conclusions

Real-time Fault Detection and Diagnosis in industrial systems have made significant progress in recent years, making it a hot research topic due to its impact on the industrial digital transformation. While only 15 studies had been published up to 2021, 14 new studies emerged in 2022 and 2023, indicating a rapid acceleration and increased interest in this field. However, there are still many opportunities to explore. Our literature review has shown challenges, such as the absence of standardized datasets and the opaque nature of Machine Learning (ML) models. Although researchers have made commendable contributions, the field lacks a unified trajectory due to fragmented research paths and a scarcity of sustained, directional efforts.

Future research into Fault Detection and Diagnosis (FDD) should prioritize several key areas. It is crucial to have standardized datasets to ensure reproducibility and allow for comparative evaluations. On the other hand, there is a need to advance the handling of unbalanced datasets and refine feature extraction techniques for temporal series data. Explainable AI (XAI) techniques tailored to industrial fault detection are essential to enhance interpretability and trustworthiness. Future studies should also focus on conducting comprehensive comparative evaluations, reducing dependency on specialized expertise, documenting real-world implementation outcomes, addressing data challenges, and enhancing real-time capabilities and integration. By focusing on these research directions, the field can advance Machine Learning-based Real-Time Fault Detection and Diagnosis methodologies, ensuring their effectiveness and applicability in industrial contexts.

The use of Real-Time Fault Detection and Diagnosis in industry depends on creating a collaborative environment between technological advancements and the practical experience of industry professionals. This entails a journey of collaboration, transparency, and continued exploration to integrate ML into industrial operations effectively, paving the way for a new era of intelligent manufacturing.

## Figures and Tables

**Figure 1 sensors-25-00060-f001:**
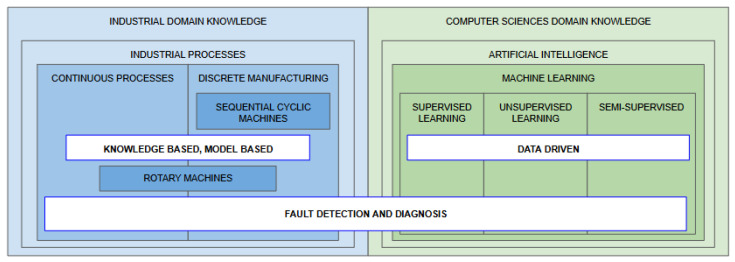
Classical and data-driven detection and diagnosis domains.

**Figure 2 sensors-25-00060-f002:**
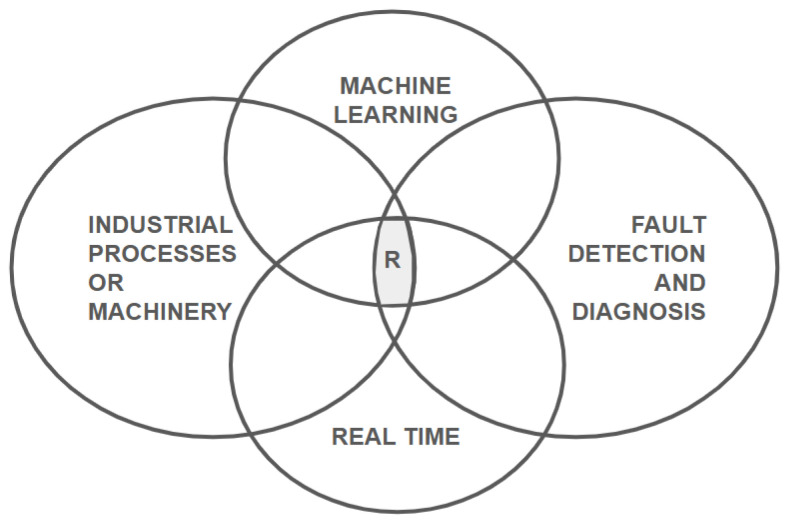
Scope of the systematic review.

**Table 1 sensors-25-00060-t001:** Combined Table of Related Systematic Literature Reviews and Summary of Key Studies in RT-FDD.

Study	Focus and Contribution	Historical Aspect	Challenges and Opportunities	Title of Review
Patton et al. [51] (1999)	Research on FDD in the 90s with Neural Networks for fuzzy rule generation.	Residual Signals	Challenge in defining thresholds for residual signals; neural fuzzy alternatives offer a solution.	Artificial Intelligence Approaches to Fault Diagnosis
Fenton et al. [52] (2001)	Fault diagnosis in electronic systems, highlighting industry adoption challenges.	Rule-based Systems	Need for solutions that suit real industrial applications and promote clear savings.	Fault Diagnosis of Electronic Systems Using Intelligent Techniques: A Review
Lo et al. [53] (2019)	ML approaches in fault diagnosis for IoT systems, suggesting hybrid methods.	Hybrid Methods	Data quality issues, need to address dynamic system variables and generalize models.	Review of Machine Learning Approaches in Fault Diagnosis applied to IoT Systems
Fernandes et al. [54] (2022)	ML for mechanical fault diagnosis in manufacturing, emphasizing interpretability and online learning.	Interpretability	Complexity of manufacturing systems, time-varying properties of processes, lack of labeled data.	Machine Learning techniques applied to mechanical fault diagnosis and fault prognosis in the context of real industrial manufacturing use-cases: a systematic literature review
Arpitha et al. [4] (2022)	ML for fault detection in semiconductor manufacturing, advocating for multivariate methods.	Multivariate vs. Univariate	Effectiveness of multivariate techniques over traditional univariate approaches in complex processes.	Machine Learning Approaches for Fault Detection in Semiconductor Manufacturing Process: A Critical Review of Recent Applications and Future Perspectives

**Table 2 sensors-25-00060-t002:** Query search parameters.

Parameter	Parameter Description	Terms
A	Application Field	(industry OR industrial OR manufacturing)
B	Application Purpose	(“outlier detection” OR “anomaly detection” OR “fault detection” OR “fault prediction” OR “condition monitoring”)
C	Computational Technique	(“machine learning” OR “artificial intelligence” OR “data mining” OR “deep learning” OR “process mining”)
D	Time Aspect	(“real-time” OR “real time”)

**Table 4 sensors-25-00060-t004:** Comparison of contributions between this review and previous reviews.

Aspect	Previous Reviews	This Review
Focus of the Review	Broad exploration of RT-FDD methods, particularly on rotary machines and specific manufacturing domains like semiconductor etching [51,52,53].	Addresses Real-Time Fault Detection and Diagnosis (RT-FDD) methods across diverse industrial contexts, including continuous and discrete manufacturing systems.
Real-World Applications	Limited coverage, often constrained to specific case studies or industrial domains, such as mechanical faults or semiconductor manufacturing [4,54].	Provides broader insights into practical challenges and gaps in real-world implementation, highlighting the lack of studies documenting the real impact of RT-FDD on industrial performance.
Dataset Availability and Standardization	Highlights issues like a lack of labeled data but does not emphasize dataset standardization or public availability [53,54].	Stresses the critical need for standardized datasets and the importance of public availability to ensure reproducibility and benchmarking.
Methodological Advances	Discusses established methods like PCA, kNN, and SVM, with limited exploration of emerging techniques [52,53,54].	Incorporates recent advances, like AutoML, Explainable AI (XAI), and hybrid approaches, addressing their potential to reduce dependency on specialized expertise.
Interdisciplinary Insights	Focuses on specific disciplines, such as electronics or mechanical systems [4,54].	Bridges industrial and computational domains by exploring complementary approaches combining physical models and data-driven methods.
Emerging Technologies	Limited focus on newer technologies like IIoT and digital twins [4].	Explores opportunities for integrating digital twins, IIoT, and edge computing into RT-FDD frameworks to maximize cross-industrial applications.
Research Gaps Identified	Limited emphasis on gaps like interpretability, scalability, and operational impact [53,54].	Highlights critical gaps, including dataset imbalance, lack of benchmarks, and absence of robust real-world implementation results.
Future Research Directions	General recommendations for improving Fault Detection and Diagnosis methods [51,53].	Detailed directions, including real-time integration, feature extraction, XAI, and addressing specific challenges in discrete manufacturing systems.
Categorization of Studies	Not explicitly categorized; insights are scattered across various topics, making it harder to identify overarching themes.	Proposes a novel categorization of studies into six distinct groups, which provides a clear framework to address challenges and identify synergies across research areas.

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
