# Peer review of "Fault Detection and Diagnosis in Industry 4.0: A Review on Challenges and Opportunities"

_sensors, 2024, doi:10.3390/s25010060_

Round 1
Reviewer 1 Report
Comments and Suggestions for Authors
1:Please elaborate on the logical relationships depicted in Figure 1.
2: Although the paper mentions some of the challenges and limitations faced by RT-FDD in industrial settings, such as technical complexity, resource requirements, etc., there is relatively little in-depth discussion of these challenges and limitations and the proposal of specific solutions.
3: Specific directions for future research or potential research gaps may not be clearly indicated in the summary section.
4: It is recommended to add the latest research in the field of deep learning, such as graph networks and KAN networks.
5. Descriptions of actual cases may be too brief and lack in-depth details and background information, such as fault composition, structure, fault data distribution, etc
Reviewer 2 Report
Comments and Suggestions for Authors
(1) The paper focuses on Industry 4.0, but it places more emphasis on the general industrial field. It is suggested that the entire manuscript be revised to better align with the central theme of Industry 4.0.
(2) In Section 2.4, the authors perform data retrieval in digital libraries based on specific keywords. However, some relevant papers that use particular machine learning (ML) or artificial intelligence (AI) techniques for diagnosis and monitoring have not been included.
(3) The paper analyzes the current research trends. However, when discussing the trend shifts in different research areas (e.g., research related to continuous processes versus discrete manufacturing processes), further exploration of the underlying reasons for these changes would be beneficial.
(4) While summarizing the selected studies, it is effective to classify the research according to different challenges (e.g., data and process complexity, early detection of anomalies or new faults, new technical methods, workshop implementation requirements, lack of labeled data, and issues of interpretability). However, the comparison and analysis of research results within the same category is somewhat limited.
(5) For each future research opportunity, it would be helpful to elaborate on specific implementation steps or research strategies. This should include an analysis of potential challenges and difficulties that might arise, as well as a discussion of the expected impact and contributions of these opportunities to both the theory and practice of real-time fault detection and diagnosis (RT-FDD), considering the practical context.
(6) It is recommended to include charts or visuals to help readers better understand key concepts, research trends, and methodologies.
Reviewer 3 Report
Comments and Suggestions for Authors
Review Fault Detection and Diagnosis in Industry 4.0
When reviewing “review” papers, I look for three elements:
1. How the authors approach the review with respect to describing and categorizing key aspects of the area of interest. This shapes the nature and conceptual approach to the review.
2. Since review outcomes are invariably about identifying and interpreting patterns, what are the groupings and labels of these patterns and are these patterns supported by the review. These are key findings of the review. The groupings are heavily at the discretion of the authors.
3. Was the review conceptualized, conducted, and interpreted in a systematic, comprehensive, and understandable manner. The process of the review and the authors’ thought and decision processes are as important as the findings.
I recommend the paper be published because there are some interesting patterns and interpretations reported and there are valuable observations, especially in the analysis and finding sections. I also believe this review adds to past reviews which was clear because the paper reviewed five past reviews on FDD. The search itself was well documented and conducted systematically covering an extensive base of literature. While I could understand the overall intent of the authors’ conceptualization of the review and the subsequent analysis, I struggled with understanding the authors’ thought process in several sections and how several sections linked to each other. These are equally important to understanding and interpreting findings because of author discretion with the content. Below, I have noted areas of valuable content but also where I struggled with richer interpretation. The expectation is that the authors can add their process and analysis context or adjust some sections accordingly.
One comment on the specifics of the search, I did look carefully at the parameters, exclusions and inclusions. These made sense but I had two lingering questions. Parameter C is intended to surface papers based on methods. I was surprised that causal reasoning, qualitative modeling, classification, feature extraction/abstraction, etc. were apparently not included even though they have seen wide use in FDD. It was also not clear how hybrid methods would be surfaced. However, hybrid approaches and quite few other methods (not listed in C) did surface in the 29 papers ultimately identified. It would therefore be useful to have some analysis of the search results themselves to explain how parameters did in fact pick up cross-industry, cross-method, hybrid approaches. This gets at your analysis of how well you felt your search parameters worked based on the papers and findings you have reported? There is work that I am aware of that did not make it through the search filters of this study. This invariably happens. However, knowing more about the analysis of your own search results and how your parameters did in fact filter results is useful. Additionally, if you did any trial-and-error tests with different parameters this would be good to know.
For item #1 of my criteria, the Previous Reviews section was to me more important and added needed clarity on the authors’ conceptual thinking than did the introduction or the other section 2 discussions. For example, there is an important observation in lines 236 – 245 to the effect that RB approaches remain dominant because of their simplicity and continued value even with shortcomings. There is therefore a need to show implementation value for using more sophisticated MB approaches. A key observation in the Lo review is about the importance of getting the data cleaned up and right. What is the one method in line 252? The summary (261-280) is a nice summary of the Fernandes review that brought out cross evaluation issues, complexity, and lack of data. The Arpitha review nicely brings out the semiconductor etching as an example of nonlinearity and high dimensionality. A reason for further emphasizing this Previous Reviews section is that it is a key reference point or baseline for this new review. The next section did make clear that this new review had built on the past reviews section. I am taking this a step further and recommending, in addition to what you have already one, that you also present your new findings relative to the Previous Reviews section to show further progress with FDD and the value of your review in this context. This was not done in your analysis sections but seemed quite doable when looking at the key results in this review in comparison to the previous reviews section.
Section 3.2 begins to address my item #2 review criteria. It is here that the 29 selected papers that reflect recent research (2018 – 2023) were analyzed and grouped into six areas. Since these six groupings are key findings, it would be useful to have the authors’ sense of how these groupings were determined when introducing section 3.2. Was there a systematic process? Table 3 is not just a summary of papers, but it is summary of the groupings and how the groups are supported by the papers. A discussion on these aspects would be useful.
I did struggle with how sections 3.2 and 4 linked together to get to section 5. For example, while the six groupings in section 3 seem to be first level findings from the 29 papers, section 4 opens by tackling a cross cut analysis of the 29 papers in time, industry, and nature of the study to identify: (1) a strong trend in using authentic data – indication of industry interest, (2) a strong cross industry segment interest, (3) use of real time data, (4) lack of explicit information in industrial use, and (5) no ability for cross evaluation/no use of benchmark datasets. Like the groupings in section 3.2, these are also valuable groupings, but they have a different flavor. I could see how the authors were probably doing analyses, but I could not link the outcomes of the two different sections.
Similarly with section 4.2, it appears that the authors went through a separate process of reviewing all 29 papers with the lens of each research question. There is some connection to the section 3 lists. It would be most useful to include some description of the process for these analyses and then be clearer how they link or build together (or be clear they were conducted as separate analyses with different lenses). Ultimately, the authors somehow took these analyses, rolled them up by prioritizing, regrouping, etc. into section 5 which lists some important and useful findings. The review is substantially strengthened if these various analyses are connected with the process used to support section 5.
Finally, I have some major comments on the introductory conceptualization and set up sections that motivate the analysis. Section 2 argues for the fundamental nature of rotary and cyclic machines. Section 2.1 overlays a context of automation systems and alarm systems and Sections 2.2 and 2.3 are intended to distinguish classical methods from data driven methods. I simply could not follow the details of the logic about how one gets to Figure 1 and then how it sets the stage for a focus on data-driven learning, i.e. cognitive, methods. It also did not lend clarity to how the authors are studying FDD in DMMs vs. batch processes, vs. continuous processes. There have been many ways to argue what is new and the value of AI/ML relative to other past methods. This description applies some different logic from many others I have seen. I only argue for being much clearer and more precise with how these arguments add up to an expected conclusion that ML is important capability. The overall intent to focus on AI/ML methods was clear enough without this set up so it didn’t affect the review per se. One other comment is that the introduction would benefit from greater descriptive precision so there is greater clarity on the terminology being used (from the literature) which will give greater clarity on how FDD is being studied in this paper. I am recommending another iteration on the introductory sections to clarify the conceptualization and motivation of the review.
In summary, my review has keyed on the process of the review so that the results can be understood better and have greater impact. I have no disagreement with any specific findings. Because this is a review the value is with the findings and their intensity and applicability to set direction. The only content item that stood out for me was with Line 69 where there is a statement about barriers to ML. I am not disagreeing with the three listed but am surprised there is no mention of resistance to sharing data, data are isolated, there are no standards or agreements for collecting and using data, and data have tended to be embedded in application code in proprietary ways.
Round 2
Reviewer 2 Report
Comments and Suggestions for Authors
The authors have revised all my suggestions. I agree to accept it.